# Screening a Spliced Leader-Based *Symbiodinium microadriaticum* cDNA Library Using the Yeast-Two Hybrid System Reveals a Hemerythrin-Like Protein as a Putative SmicRACK1 Ligand

**DOI:** 10.3390/microorganisms9040791

**Published:** 2021-04-09

**Authors:** Tania Islas-Flores, Edgardo Galán-Vásquez, Marco A. Villanueva

**Affiliations:** 1Unidad Académica de Sistemas Arrecifales, Instituto de Ciencias del Mar y Limnología, Universidad Nacional Autónoma de México, UNAM, Prolongación Avenida Niños Héroes S/N, Puerto Morelos, Quintana Roo 77580, México; 2Departamento de Ingeniería de Sistemas Computacionales y Automatización, Instituto de Investigación en Matemáticas Aplicadas y en Sistemas, Universidad Nacional Autónoma de México, UNAM, Circuito Escolar 3000, Ciudad Universitaria, Ciudad de México CP 04510, México; edgardo.galan@iimas.unam.mx

**Keywords:** cDNA library, coral reefs, Hemerythrin-like protein, RACK1, spliced leader, *Symbiodinium*, yeast two-hybrid

## Abstract

The dinoflagellate Symbiodiniaceae family plays a central role in the health of the coral reef ecosystem via the symbiosis that establishes with its inhabiting cnidarians and supports the host metabolism. In the last few decades, coral reefs have been threatened by pollution and rising temperatures which have led to coral loss. These events have raised interest in studying Symbiodiniaceae and their hosts; however, progress in understanding their metabolism, signal transduction pathways, and physiology in general, has been slow because dinoflagellates present peculiar characteristics. We took advantage of one of these peculiarities; namely, the post-transcriptional addition of a Dino Spliced Leader (Dino-SL) to the 5′ end of the nuclear mRNAs, and used it to generate cDNA libraries from *Symbiodinium microadriaticum*. We compared sequences from two Yeast-Two Hybrid System cDNA Libraries, one based on the Dino-SL sequence, and the other based on the SMART technology (Switching Mechanism at 5′ end of RNA Transcript) which exploits the template switching function of the reverse transcriptase. Upon comparison of the performance of both libraries, we obtained a significantly higher yield, number and length of sequences, number of transcripts, and better 5′ representation from the Dino-SL based library than from the SMART library. In addition, we confirmed that the cDNAs from the Dino-SL library were adequately expressed in the yeast cells used for the Yeast-Two Hybrid System which resulted in successful screening for putative SmicRACK1 ligands, which yielded a putative hemerythrin-like protein.

## 1. Introduction

Symbiodiniaceae are highly diverse dinoflagellate algae that establish symbiosis with cnidarians and marine organisms of diverse phyla including corals. Coral reefs shelter a highly biodiverse ecosystem, which depends on the functional symbiosis with dinoflagellate members of such a family [1,2]. The dinoflagellates reside in symbiosomal membranes in the host gastrodermal cells, where they carry out the photosynthetic process that fuels the productivity and diversity of the coral reef ecosystems. In the past decades, coral reefs have suffered large areal losses due to coral death caused by diseases and coral bleaching, the latter being a cause of the symbiosis breakdown; however, the mechanisms underlying the metabolism, symbiont selection and establishment, maintenance, and disruption of the symbiosis, are still poorly understood. The slow progress in the study of cnidarian–dinoflagellate symbiosis is a consequence of the scarcely available molecular and functional genomic tools applied to corals and Symbiodiniaceae. To date, it has not been possible to obtain silenced, mutated, or knockout stable lines of Symbiodiniaceae to carry out integral functional genomic studies that may shed light on these pathways [3,4].

Symbiodiniaceae present genomic features that make them particularly interesting: (a) they possess some of the largest nuclear genomes among eukaryotes; and (b) their DNA is arranged on permanently condensed liquid-crystalline chromosomes. This way of organization has been proposed as a third state of chromosomal folding besides the nucleosomal and the supercoiled circular arrangements in eukaryotes and bacteria [5,6]. A third and significant genomic feature is that their transcripts contain a 22-nt conserved sequence called Dino Spliced Leader (Dino-SL), DCCGUAGCCAUUUUGGCUCAAG (D = U, A or G), which is post-transcriptionally added to the 5′ end of all their nuclear mRNAs [7]. The spliced leader gene has an intronic region with unknown function, and an exonic region which is transferred (*trans*-spliced) to the 5′ pre-mRNAs [8]. The *trans*-splicing of the SL sequence is part of an mRNA processing mechanism, in which a small fragment of a non-coding RNA is transferred to a splice acceptor at the 5′-UTR of pre mRNAs [9]. The Dino-SL sequence has proved to be useful as a molecular tag that allows us to: a) specifically select dinoflagellate mRNA from mixed microbial samples [10]; (b) construct cDNA libraries from environmental samples using a Dino-SL based approach with high specificity [10,11]; and (c) identify any dinoflagellate gene [12]. These approaches have been useful to obtain specific cDNA libraries from dinoflagellates; however, it has not been explored whether they can be used for accomplishing high-quality cDNA libraries that would allow to collect full-length cDNA clones efficiently [13]. The actual methodology to construct a cDNA library involves the synthesis of second-strand cDNA, which can produce artifacts, including the loss of information at the 5′ and 3′ ends of mRNAs where gene regulation information is encoded [14]; however, this step is necessary to avoid the loss of less-abundant transcripts.

The Dino-SL is a convenient tag that allows to prime the 5′ end of the dinoflagellate mRNA, in addition to the dT at the 3′ end. Therefore, it is expected that, when using the Dino-SL tag, cDNA libraries representing a higher number of complete transcripts as well as a higher representation of less-abundant transcripts, will be obtained. The result would be a molecular tool allowing to analyze the sequencing data obtained from them, with more reliability than from conventionally constructed libraries. Furthermore, it opens the opportunity to generate cDNA libraries for applications like the Yeast-Two Hybrid System (Y2HS) that can be used to identify ligands of proteins of interest.

The Y2HS is one of the most widely used methods for in vivo identification of interacting proteins, confirming interactions between two known proteins, and/or mapping interacting domains; in addition, it has been used to successfully map interaction networks on a large scale [15]. In this technique, the interaction between two proteins of interest is detected via the reconstitution of a transcription factor and the subsequent activation of reporter genes under the control of a transcription factor [16]. It exploits the fact that the DNA-binding domain (BD) of GAL4 is incapable of activating transcription unless physically associated with an activating domain (AD) [17]. In a two-hybrid assay, a protein of interest is fused to a DNA binding domain (BD), then transfected to a yeast strain; this construct is called the “bait”. In order to be useful, the “bait” must not be able to activate the transcription of the reporter(s) on its own, so that a cDNA library that is fused to the activation domain (AD) called the “prey” can be screened. Thus, when “prey” library clones express proteins capable of interacting with the “bait”, they can be identified by their ability to activate the reporter(s). Although this system presents an opportunity to identify or confirm protein–protein interactions, it is not a failsafe approach [17]. Success in the identification of ligands by the Y2HS depends on several factors that can negatively affect the detection of protein–protein interactions. For example, the use of chimeras is a concern because it can change the protein (“bait” and/or “prey”) conformations and thus disrupt their interaction sites; another drawback is the use of yeast as a host to express the cDNA library of heterologous systems, which implies that some required post-translational modifications for protein–protein interactions could be incorrectly carried out or not occurring at all. Finally, cDNA libraries with underestimated or incomplete transcripts could lead, in turn, to incomplete, untranslated (incorrect frame), or low amount of “prey” proteins. The Dino-SL based cDNA library approach could overcome all those problems. In this work, in an attempt to obtain a library with higher complexity than the conventionally generated SMART (Switching Mechanism at 5′ end of RNA Transcript) libraries, we sought to generate a cDNA library of *Symbiodinium microadriaticum* ssp. *microadriaticum* using the Dino-SL strategy (DINO library) to amplify the dinoflagellate cDNA and clone it into the pGADT7-rec plasmid. Then, to confirm that with this approach we could indeed obtain a longer and higher number of transcripts, we compared it to a conventionally generated cDNA library (SMART library). The DINO library allowed us to obtain 54% more transcripts with a ≥5000 bp length, than with the SMART library. In order to validate the DINO library as a tool for the Y2HS, we used the Receptor for Activated C Kinase 1 from *S. microadriaticum* (SmicRACK1) as bait to screen the library for prey transcripts and obtained a sequence that upon BLAST [18] matched a hemerythrin-like protein as a putative SmicRACK1 ligand. This was further confirmed by direct activation of the reporter gene by both sequences in a second Y2H assay. Thus, the Dino-SL strategy provides a molecular tool to obtain high throughput cDNA libraries that can be used as expression libraries for molecular approaches.

## 2. Materials and Methods

### 2.1. Cell Lines and Cultures

Aseptic cultures of the photosynthetic dinoflagellate *Symbiodinium microadriaticum* ssp. *microadriaticum* (MAC-CassKB8), originally isolated from the jellyfish *Cassiopea xamachana*, and kindly donated by Dr. Mary Alice Coffroth (State University of New York at Buffalo), were routinely maintained as in vitro culture in ASP-8A medium [19] under standard photoperiod cycles of 12 h light/dark with a light regime of 80 μmoles quanta/m^2^ s, at 25 °C. The yeast strain Y187 was used as host for both “prey” cDNA libraries. This strain is stored in our laboratory as glycerol stocks at −80 °C, and also maintained on PDA medium at 4 °C.

### 2.2. Total RNA Isolation, cDNA Synthesis and cDNA Amplification

Total RNA from exponentially growing *S. microadriaticum* cultures (1 × 10^5^ cells/mL) was extracted as follows. Cultures were harvested by centrifugation at 1400× *g* for 5 min, then the pellet was suspended with TRI REAGENT (Sigma, St. Louis, MO, USA) and vigorously shaken at 48 rpm with glass beads (425–600 µm; Sigma) in a bead beater (Biospec, Bartlesville, OK, USA) for 2–3 min at 4 °C; total RNA was extracted according to the manufacturer’s instructions. The RNA was further purified with the RNA CLEANUP Kit (Qiagen, Valencia, CA, USA), and mRNA was purified with the OLIGOTEX kit (Qiagen, Valencia, CA, USA) according to the protocol in the accompanying user manual. All the RNA samples were treated with RNase-free DNase I (Thermo Fisher Scientific, Waltham, MA, USA) to eliminate DNA. The integrity of total RNA was analyzed by 1.2% denaturing agarose gel electrophoresis, followed by GELRED staining (Biotium, Fremont, CA, USA). The concentration and purity of the total RNA were determined with a SmartSpec 3000 spectrophotometer (BioRad, Hercules, CA, USA) at 260 and 280 nm, respectively. The cDNA synthesis was made with the Matchmaker Library Construction kit (Clontech Laboratories Inc., Mountain View, CA, USA), according to the manufacturer’s instructions. Two tubes with 0.5–1 µg of mRNA were mixed with 1 µL 10 µM CDSIII oligo (5′ ATTCTAGAGGCCGAGGCGGCCGACATG-d(T)30VN 3′) and 2 µL of nuclease-free water. The tubes were incubated at 72 °C for 5 min and then quickly cooled on ice before their content was added to two tubes containing 2 µL of 5X first strand buffer, 1 µL 0.1 M DTT, 1 µL 10 mM dNTP’s mix, and 1 µL of the SMART Moloney murine leukemia virus reverse transcriptase (M-MLV RT). Then, the mixtures were incubated as follows: 42 °C for 10 min, followed by 1 µL of SMARTIII oligo (5′ AAGCAGTGGTATCAACGCAGAGTGGCCATTATGGCCGGG 3′) to only one tube; the incubation continued for 1 h at 42 °C; 75 °C for 10 min; and 23 °C for 5 min. At this point, 1 µL of RNAse H was added to both tubes; finally, the tubes were incubated at 37 °C for 20 min. 

Long distance-polymerase chain reaction (LD-PCR) was carried out with the ADVANTAGE 2 POLYMERASE MIX (Clontech) following the manufacturer’s instructions. For each cDNA (synthesized with or without the SMARTIII oligo), two 100 µL reactions were carried out. The reactions consisted of a mixture of 10 µL 10X Advantage 2 PCR Buffer, 2 µL 50X dNTPs mix, 2 µL 10 µM 3′ PCR oligo (5′ GTATCGATGCCCACCCTCTAGAGGCCGAGGCGGCCGACA 3′), 10 µL 10X fusion solution, 2 µL 50X Advantage 2 polymerase mix, and 70 µL nuclease-free water along with: (a) 2 µL first strand cDNA synthesized with SMARTIII oligo and 2 µL 10 µM 5′ PCR oligo (5′ TTCCACCCAAGCAGTGGTATCAACGCAGAGTGG 3′); or (b) 2 µL first strand cDNA synthesized without SMARTIII oligo and 2 µL 10 µM DINO-SL oligo (5′ TTCCACCCAAGCAGTGGTATCAACGCAGAGTGGCCATTATGGCCCCGTAGCCATTTTGGCTCAAG 3′). Then, the reactions were subjected to the following program: step 1, 94 °C, 30 s; step 2, 94 °C, 10 s; step 3, 68 °C, 6 min; 25 cycles from step 2–3. To confirm that the amplification took place (which should be seen as a smear on the gel), 7 µL of each reaction were analyzed on a 1.2% agarose gel subsequently stained with GelRed (Biotium, Fremont, CA, USA). The PCR products were purified with CHROMA SPIN^TM^ +TE-400 columns to discard DNA molecules of <200 pb.

### 2.3. Generation of cDNA Libraries

Two to five µg from each ds cDNA pool were mixed with 3 µg of *Sma*I linearized pGADT7-rec vector, and Y187 competent cells were transformed at library scale using the Yeastmaker Yeast Transformation System 2 (Clontech) according to the protocol in the accompanying user manual. The SMART and Dino-SL (DINO) libraries had 7.2 × 10^8^ and 2.43 × 10^9^ independent clones, respectively, which was an indication of the library complexity (which should be no less than 1 × 10^6^ independent clones).

### 2.4. Sequencing of Libraries

To determine that the Dino-SL based approach improves the cDNA library with respect to the use of the regular methodology, we decided to sequence both Y2HS (DINO and SMART) libraries. A 1 mL aliquot from each library was thawed in a water bath at 25 °C. The plasmids from the libraries were purified with the ZYMOPREP YEAST PLASMID MINIPREP II kit (Zymo Research, Irvine, CA, USA). The libraries were amplified by LD-PCR with the ADVANTAGE 2 POLYMERASE MIX (Clontech) using the AD-Screen Fwd oligo (5′ CTATTCGATGATGAAGATACCCCACCAAACCCA 3′), and the AD-Screen Rv oligo (5′ GTGAACTTGCGGGGTTTTTCAGTATCTACGATT 3′) under the same conditions of cDNA amplification. The PCR products were sent for sequencing to the Mass Sequencing and Bioinformatics facility of the Institute of Biotechnology at UNAM. Sequencing was carried out by Illumina (2 × 75 cycles, 10 million reads).

### 2.5. Bioinformatic Analysis

Read quality control and cleaning were carried out using Fastp (v0.19.5), which can perform quality control, adapter trimming, quality filtering, per-read quality pruning and many other operations with a single scan of the FASTQ data. Default parameters were used [20]. The de novo assembly was carried out using Trinity (v2.8.4) [21], from the sequences free of: adapters, overrepresented sequences, low quality and low complexity. Then, TransDecoder was used to identify candidate coding regions from the assembled transcripts (v5.5.0). The candidate coding regions were used for BLAST analysis [18] against the draft of *S. microadriaticum* genome [22,23], and homologs were accepted if they had an e-value < 1 × 10^−5^, sequence identity > 90%, and alignment length > 90% of the individual proteins. The presence of conserved domains in the assembled transcripts was identified and annotated using HMMER [24]. KEGG pathway analysis of the candidate coding regions were performed using GhostKOALA [25].

Finally, the Kozak sequence for *Symbiodinium* (RCCATGGCN) was mapped using the DNA-pattern program which allows to search all occurrences of a pattern within DNA sequence [26]. The searches were done only in the direct strands, machining positions can be calculated either relative to the sequence start and 1 substitution were allowed.

### 2.6. Yeast-Two Hybrid System (Y2HS) Screening

In order to probe whether the DINO library was suitable to screen for protein interactions, we made a screen using the AH109 yeast strain harboring the bait construction pGBKT7-*SmicRACK1* as bait, and the DINO library as the prey in the Y187 yeast strain. The screening was carried out with the MATCHMAKER GAL4 Two-Hybrid System 3 kit following the instructions of the manufacturer (Clontech).

## 3. Results

### 3.1. The DINO Library Has Increased Yields and Higher Representation

The yield of the library is an important indicator of the transcript representation. Because the cDNA content quality could affect the performance of the libraries, we determined and compared the yield of each library to confirm differences in their complexity. Two *S. microadriaticum* full-length cDNA libraries were generated, one based on the SMARTIII oligo and the other one based on the Dino-SL tag. Both libraries had similar yield; the DINO library had 1.8 × 10^8^ ufc/mL while the SMART library 1.6 × 10^8^ ufc/mL. However, the number of independent clones was higher in the DINO (2.43 × 10^9^) than in the SMART library (7.2 × 10^8^) (Table 1). This indicates a favorable representation of transcripts in both libraries albeit with a better representation in the Dino-SL based library.

### 3.2. The Dino Spliced Leader (DINO) Library Presents Higher Transcript Identification

Because the Dino-SL sequence and a dT-based oligo were used as forward and reverse primers, respectively, to amplify the cDNA for the DINO library, it was expected that it was represented by a higher number of complete transcripts than the SMART library. The highest number of reads filtered with Fastp (v0.19.5) was obtained from the DINO library (35,311,534) compared with the number of reads obtained from the SMART library (17,715,026; Table 2); the sequenced reads had a mean length of 70 and 69 bp for the DINO and SMART libraries, respectively. After the de novo assembly, the sequences >1000 bp for the DINO library were 1.39-fold higher in the DINO library (1,168,986) than those in the SMART library (840,010) (Figure 1). Similarly, the sequences harboring >5000 bp were ~2.2-fold higher in the DINO library (89,797) than those in the SMART library (40,817) (Figure 1). The calculated N50 scaffold for the DINO library was 5671, while that for the SMART library was 3611 (Table 2). The de novo assembly produced 4760 transcripts from the DINO library, and 3611 transcripts from the SMART library, respectively (Table 2). The transcripts were used for the detection of coding sequences (CDS) and prediction of proteins resulting in the identification of 3117 CDS and 3117 proteins in the DINO library, and 2025 CDS and 2025 proteins in the SMART library (Table 2). These data indicated that the cDNA libraries based on Dino-SL allowed us to obtain a significant information gain.

### 3.3. Higher 5′ Representation in the DINO Library 

In most known cDNA library databases, several sequences are truncated towards the 5′ end either partly or completely. This results in sequences lacking the initiation site and, therefore, upon further processing, they result in incorrect ORFs. Thus, in order to identify the translation initiation in unique sequences for each library, we analyzed the Kozak sequence for *Symbiodinium* RCCATGGCN described in Zhang et al. [11].

Among the 1422 unique genes in the DINO library, 1128 contained the Kozak sequence which represents 78.22% of the total. On the other hand, from the 1070 unique genes of the SMART library, 776 contained the Kozak sequence which represents 72.52% (Figure 2a).

### 3.4. Identification of Unique and Shared Orthologs in Assemblies

To identify the unique proteins in the libraries, the orthologous proteins between them, and those from the draft of the *S. microadriaticum* genome [22,23] were identified. A number of 1422 unique proteins was identified from the DINO library and 1070 unique proteins from the SMART library, respectively, including 446 proteins shared between the two libraries (Figure 2b). Upon functional annotation using KEGG for each of the libraries, the highest proportion of the proteins in the DINO library was found to be related to genetic information processing (256 genes), carbohydrate metabolism (65), signaling and cellular processes (37), and environmental information processing (35). In parallel, these proteins were found to be related to different metabolic pathways which included those involved in carbohydrate metabolism such as glycolysis/gluconeogenesis (13), glyoxylate and dicarboxylate metabolism (8), butanoate metabolism (7), the citric acid cycle, (6), pentose phosphate pathways (6) and those involved in energy metabolism such as photosynthesis (16), methane metabolism (10), and oxidative phosphorylation (8). Others were related to carbon fixation in photosynthetic organisms (8); lipid metabolism as fatty acid degradation (9), amino acid metabolism such as glycine, serine, and threonine metabolism (7), and cysteine and methionine metabolism (6), among others. On the other hand, the proteins in the SMART library were related to genetic information processing (179 genes), carbohydrate metabolism (72), signaling and cellular processes (37), and energy metabolism (30). Furthermore, these proteins were associated to different metabolic pathways which included carbohydrate metabolism as glycolysis/gluconeogenesis (12), starch and sucrose metabolism (9), the citrate cycle, (5), and pentose phosphate pathways (6); others were related to energy metabolism such as photosynthesis (14), carbon fixation in photosynthetic organisms (8), methane metabolism (6); and lipid metabolism such as fatty acid degradation (7); and finally, others related to amino acid metabolism such as valine, leucine and isoleucine degradation (8), and alanine, aspartate and glutamate metabolism (6), among others (Figure 3, Appendix A).

### 3.5. Identification of the Receptor for Activated C Kinase 1 from S. microadriaticum (SmicRACK1) Ligands by Y2HS Using the DINO Library

One of the main concerns for using the Y2HS to identify protein-protein interactions is that with the use of heterologous systems like this one, the correct expression of heterologous proteins can be affected. Due to this, we made a Y2HS screen of putative SmicRACK1 ligands to prove that the DINO library can be suitable to identify protein-protein interactions. We sought to identify ligands of the Receptor for Activated C Kinase 1 from *S. microadriaticum* (SmicRACK1) [27]. Our Y2HS screen identified several putative SmicRACK1 ligands (Appendix A) whose sequences, in general, contained the Dino-SL sequence, a short 5′ UTR and a 3′ UTR. One of these was identified as belonging to the hemerythrin-like superfamily (Figure 4) and presented again the Dino-SL sequence followed by a 5′ UTR (44 bp), a CDS with start and stop codons, and a 3′ UTR (Figure 4). A second Y2H assay using each sequence as bait and prey, correspondingly, resulted in the activation of the reporter gene and confirmed the interaction between these two proteins (Appendix A). This indicated that the Dino-SL tag in dinoflagellates is a useful tool to generate cDNA libraries not only for sequencing, but also to identify ligands by Y2HS.

## 4. Discussion

The mechanisms underlying Symbiodiniaceae–cnidarian mutualistic symbiosis are still poorly understood even though it is a fundamental association for the survival and biodiversity of coral reefs. This association is also sensitive to environmental stress [28], particularly to the rise of temperature and light which causes its breakdown. Analogous to the former, there has been slow progress in the study of the molecular mechanisms involved in this disruption, mainly due to an absence of appropriate techniques to study the proteins, functions, and signaling pathways that work to maintain the symbiotic association. In order to develop a tool that could aid in the identification of molecular mechanisms that govern the life cycle and the Symbiodiniaceae–cnidarian symbiosis, we took advantage of the Dino-SL sequence found in the Symbiodiniaceae transcripts to generate a cDNA library from *S. microadriaticum* and analyze its performance and suitability in the Y2HS, compared to a SMART-generated library. In dinoflagellates, the spliced leader trans-splicing process transfers a Dino-SL sequence to the 5′ end of independent immature transcripts [7,8]. On the other hand, the SMART technology takes advantage of the reverse transcriptase ability to switch templates, reducing the occurrence of incomplete cDNA synthesis. However, the latter does not discriminate between full length and truncated transcripts [29], which results in a decrease of full-length cDNAs. The Y2HS is one of the molecular tools used to find in vivo novel protein–protein interactions by the reconstitution of a transcription factor that activates reporter genes. This approach has been used in assays with several cDNA libraries from different organisms, but not Symbiodiniaceae. Thus, we sought to explore the possible advantages and application of a DINO library from *S. microadriaticum* to identify putative SmicRACK1 ligands by Y2HS. 

Our comparative analysis indicated that a cDNA library generated with the Dino-SL sequence was most suitable to obtain a higher recovery of transcript ORFs which, in addition, included more sequence of the 5′ end UTRs compared to that from the SMART library. In addition, the DINO library presented a higher number of independent clones (2.43 × 10^9^) with respect to the SMART library (7.2 × 10^8^) (Table 1). The number of independent clones is indicative of the library complexity, which implies that the DINO library had better representation of transcripts. Furthermore, the complexity of the libraries was confirmed by the total reads obtained after sequencing. Up to double total reads were obtained for the DINO library (35,311,434 M), compared with the SMART library (17,715,026 M) (Table 2). This improvement in the library yield results in an enrichment of transcripts without the use of expensive kits or complicated procedures and only an oligonucleotide with the Dino-SL sequence to replace the SMART technology is required. In practical terms for the Y2HS assay, this also results in sequences of the “prey” library without incorrect ORFs and capable of expressing complete proteins, thus elevating the probability of obtaining reporter-activated colonies through interacting ligands during the screening process.

It is important to point out that the sequences <1000 bp retrieved after the de novo assembly were present in more abundance in both libraries (70.6% and 71.13% for the DINO and SMART libraries, respectively) (Figure 1). This indicates the presence of truncated cDNAs in the two libraries. Likewise, the percent of sequences between 1000 and 5000 bp, and >5000 bp remained consistent in both libraries (Figure 1), which indicates that both methodologies have similar performance. Nevertheless, the libraries differ in the yield of recovered sequences because of the use of the Dino-SL instead of the SMART technology; the latter supposed to select for complete transcripts. The identified unique coding sequences were also higher for the DINO library (1422) than for the SMART library (1070); and the libraries shared only 446 of the coding sequences (Figure 2b). This low number of shared sequences is very likely the result of the SMART library harboring more incomplete CDS, therefore, escaping detection in the analyses. The number of coding sequences identified for the DINO library was lower than expected considering the high number of assembled sequences obtained; however, it is also a result of the multicopy genome of Symbiodiniaceae, whose genome presents multiple gene copies from gene duplication; i.e., the genome of *S. goreaui* displayed a genome-fragment duplication of 15.31% (5498 of 35,913) of the predicted genes [30]. This is also evident from their reported genomes, which have revealed a significant portion of genes arranged in tandem [31].

These data suggest that the increased number of sequences of the Dino-SL based library is likely due to a diminished loss of transcripts and 5′ ends, thus resulting in a better representation than in the SMART library. As the Dino-SL oligo selectively amplifies dT oligo-synthesized cDNA harboring the Dino-SL sequence, the DINO library should be enriched for sequences that contain the translation start site and upstream. It is well known that dinoflagellates such as *S. kawagutii*, use strong Kozak motifs as consensus sequences of the translation initiation site [11,32]. Thus, we examined the presence of the AUG start codon flanked by the Kozak sequence in the unique gene sequences of both libraries. There was a prevalence of a higher number of genes that presented this translation start site in the DINO library (78.22%, 1128 from 1422 genes) than those in the SMART library (72.52%, 776 from 1070 genes (Figure 2a). This indicates that by using the Dino-SL strategy, a greater coverage of the 5′ end, including the translation start site, which is an indispensable requirement for an expression library, is obtained.

The functional annotation of the libraries shows a high coincidence between both libraries; however, there were no proteins related to the categories metabolism of terpenoids and polyketides in the DINO library. On the other hand, there were no proteins related to the category biosynthesis of other secondary metabolites in the SMART library (Appendix A). In addition, the heatmap of metabolic pathways indicates a similar pattern in both libraries (Figure 3), but whose differences are the result of the difference in the methodologies (from generation of the libraries to sequencing). Thus, the differences in the methodologies did not have an important effect on the pool of transcripts or their abundance. 

The spliced leader trans-splicing phenomenon is not particular for dinoflagellates. It was first described in trypanosomes [33,34,35], and later it was also identified in *Euglena* [36], nematodes [37], platyhelminthes [38], chordates [39,40,41,42], cnidarians [43], rotifers [44], dinoflagellates [7,45,46], porifers [47], ctenophors [47,48], and chaetognaths [49]. However, although there are several spliced leader consensus sequences in most of these organisms and the trans-splicing process does not occur in all the mRNAs, this feature is only useful in cases of a unique consensus sequence occurring in the whole mRNA population, i.e., rotifers, chordates and dinoflagellates [8].

SmicRACK1 belongs to the RACK1 and WD-40 protein families [27], it is expressed ubiquitously in eukaryotes, and is related to multiple cellular processes due to its scaffolding properties that enable it to interact with multiple ligands at the same time [50]. We conducted a Y2HS screen to identify SmicRACK1 ligands and to confirm the functionality of the DINO library. We identified several putative SmicRACK1 ligand sequences that contained the recognizable Dino-SL sequence, 5′ UTR, CDS and 3′ UTR. One of the ligand CDS encoded a Hemerythrin-like superfamily protein (Figure 4). Hemerythrin proteins belong to four family members of oxygen-carrier non-heme di-iron bonding proteins (hemoglobin, hemerythrin and two non-homologous families of arthropodan and molluscan hemocyanins) [51,52]. Hemerythrin proteins have been identified in invertebrates (sipunculans, polychaetes, priapulids and brachiopods) [53], and in 367 bacterial, 21 archaeal and 4 eukaryotic genomes [51]. Their functions are involved in signal transduction, phosphorelay regulation, abiotic resistance, and protein binding [51,52]. The function of this protein in *S. microadriaticum* is yet to be determined, but its characteristics point to an involvement in oxygen and iron storage and transport. Confirmation of its function will allow the physiological processes of *S. microadriaticum* in which SmicRACK1 participates to be identified.

## 5. Conclusions

Our comparative analysis clearly indicates that using the Dino-SL sequence found in the transcripts of *S. microadriaticum* to generate cDNA libraries allows a higher transcript recovery to be obtained when compared to a SMART technology-generated library. Furthermore, a combined strategy using both types of library will increase the probability of obtaining a more comprehensive coverage of transcripts to enhance the corresponding application to which the libraries are to be used. This molecular tool will enable progress in understanding the molecular events underlying the biology, metabolism and symbiosis of Symbiodiniaceae.

## Figures and Tables

**Figure 1 microorganisms-09-00791-f001:**
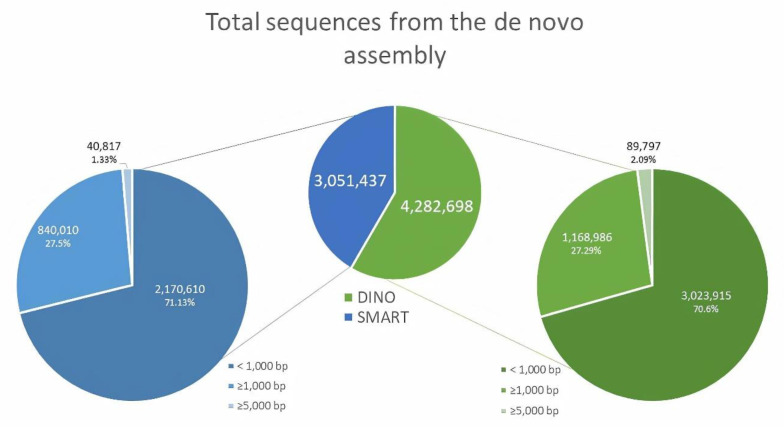
Number of sequences from the de novo assembly. The raw sequences were de novo assembled using Trinity (v2.8.4). The total number of sequences for the DINO (Dino Spliced Leader) library was 4,282,698; of these, 3,023,915 sequences were <1000 bp; 1,168,986 were from 1000 to 5000 bp; and 89,797 were >5000 bp. The total number of sequences for the SMART (Switching Mechanism at 5′ end of RNA Transcript) library was 3,051,437, distributed as follows: 2,170,610 were <1000 bp; 840,010 sequences between 1000 to 5000 bp; and 40,817 sequences >5000 bp.

**Figure 2 microorganisms-09-00791-f002:**
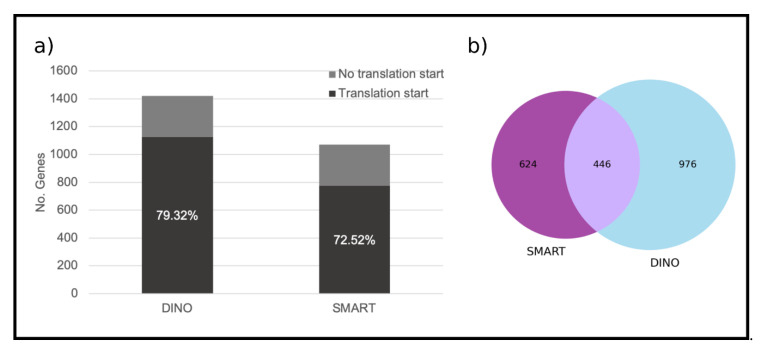
Total number of genes present in the libraries. (**a**) Percentage of sequences showing the Kozak sequence; (**b**) Venn diagram of the genes shared between both libraries.

**Figure 3 microorganisms-09-00791-f003:**
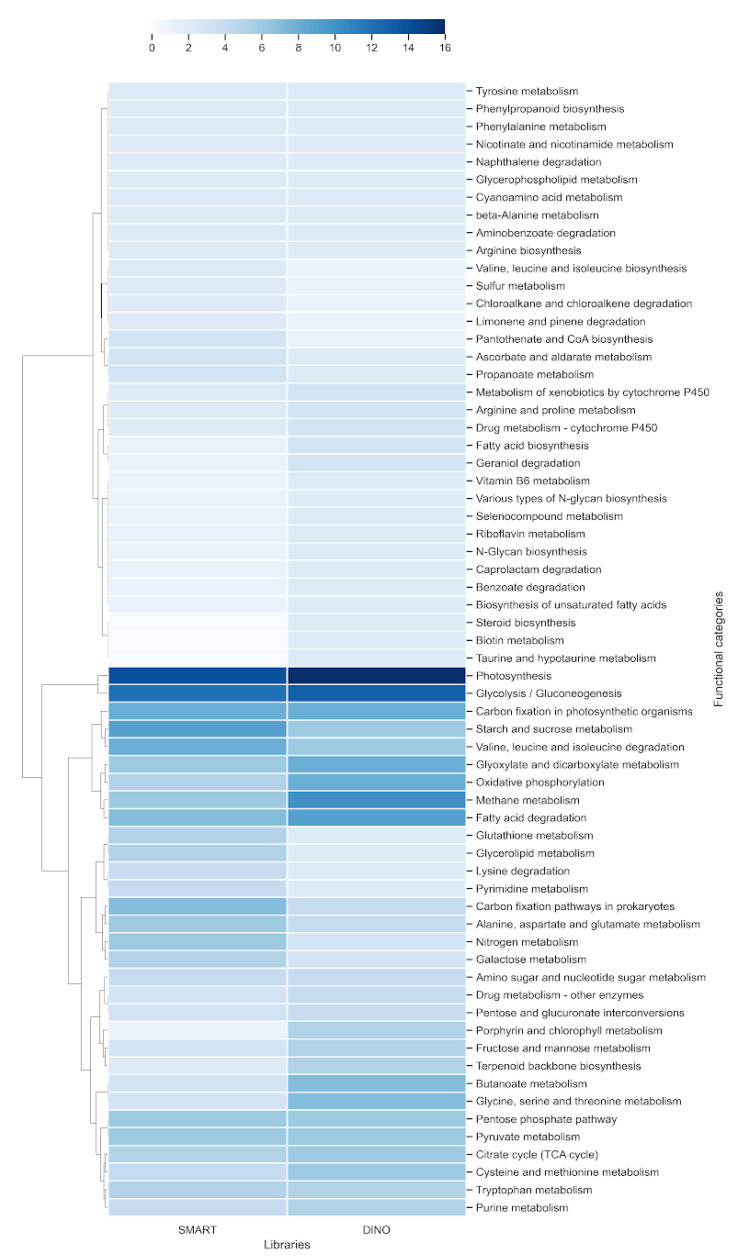
Heatmap of metabolic pathways associated with each library. A hierarchical cluster based on Euclidean distance measure and Ward’s method for linkage analysis was carried out. Each row represents the metabolic pathways term.

**Figure 4 microorganisms-09-00791-f004:**
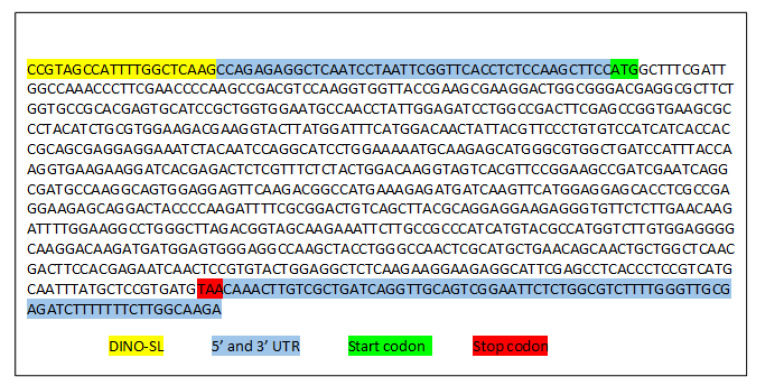
The sequence of the putative Receptor for Activated C Kinase 1 from *S. microadriaticum* (SmicRACK1) ligand, hemerythrin-like protein identified by the Y2HS. The obtained sequence presents all the elements of a full-length transcript. The ligand sequence was identified after carrying out the Y2HS screening of the cDNA DINO library. The Dino-SL sequence is shown in yellow; the 5′ and 3′ UTR’s in blue; the START and STOP codons are shown in green and red, respectively.

**Table 1 microorganisms-09-00791-t001:** Obtained yields from the DINO (Dino Spliced Leader) and SMART (Switching Mechanism at 5′ end of RNA Transcript) libraries.

	Recommended Minimum	DINO	SMART
Density cells/mL	>2 × 10^7^	6 × 10^8^	5.9 × 10^8^
UFC/mL	>1 × 10^7^	1.8 × 10^8^	1.6 × 10^8^
Independent clones n:	>1 × 10^6^	2.43 × 10^9^	7.2 × 10^8^

**Table 2 microorganisms-09-00791-t002:** Read and assembly statistics for DINO and SMART libraries. Overview of the sequencing data, assembly, clustering, and annotation.

	DINO Library	SMART Library
Total reads (FastP v0.19.5)	35.311534 M ^1^	17.715026 M
Total bases	2.479454 Gb ^2^	1.236204 Gb
Transcripts (Trinity)	4760	3611
Total length contigs after TransDecoder		
N50 ^3^	5671	4840
N75 ^4^	4270	3534
GC content (%)	40.93	35.78
CDS ^5^	3117	2025
Proteins	3117	2025
Orthologs	1422	1070

^1^ M: millions; ^2^ Gb: gigabases; ^3^ N50: sequence length of the shortest contig at 50% of total genome length; ^4^ N75: same as N50 but length of shortest contig at 75%; ^5^ CDS: coding sequence.

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
