# Peer review of "Screening a Spliced Leader-Based *Symbiodinium microadriaticum* cDNA Library Using the Yeast-Two Hybrid System Reveals a Hemerythrin-Like Protein as a Putative SmicRACK1 Ligand"

_microorganisms, 2021, doi:10.3390/microorganisms9040791_

Round 1

Reviewer 1 Report

This interesting work is devoted to the generation and comparison of two Yeast TwoHybrid System cDNA libraries from the Symbiodinium microadriaticum based on the Dino-SL sequence and the SMART technology. Authors have demonstrated that the Dino-SL based library provides a higher yield, number, and length of sequences.

The manuscript is well written and illustrated.

There are several Comments.

  1. Please, indicate the source of the dinoflagellate Symbiodinium microadriaticum microadriaticum (MAC‐CassKB8) strain.
  2. Please, provide the references on growth mediums that have been used in this study.
  3. “…A number of 1,422 unique proteins were identified from the DINO library and 1,070 unique proteins from the SMART library, respectively, including 446 proteins shared between the two libraries (Fig. 2b)” (page 7).

 How can the authors explain that two libraries shared only 446 proteins?

It means that both molecular technologies (Dino-SL strategy (DINO library) and cDNA library (SMART library)) have to be applied to get more proteins (1,422+1,070-446=2,046). Isn't it?

Author Response

Dear Reviewer No. 1:

Please find below our response to your queries:

Q. Please, indicate the source of the dinoflagellate Symbiodinium microadriaticum microadriaticum (MAC‐CassKB8) strain.

A. The information was added on lines 122-124: "...originally isolated from the jellyfish Cassiopea xamachana, and kindly donated by Dr. Mary Alice Coffroth (State University of New York at Buffalo)..."

Q. Please, provide the references on growth mediums that have been used in this study.

A. The reference No. 19 (Blank, 1987) was added on line 124.

Q. “…A number of 1,422 unique proteins were identified from the DINO library and 1,070 unique proteins from the SMART library, respectively, including 446 proteins shared between the two libraries (Fig. 2b)” (page 7).

 How can the authors explain that two libraries shared only 446 proteins?

A. The explanation has been added in the discussion on lines 371-373: "This low number of shared sequences is very likely the result of the SMART library harboring more incomplete CDS, therefore escaping detection in the analyses."

Q. It means that both molecular technologies (Dino-SL strategy (DINO library) and cDNA library (SMART library)) have to be applied to get more proteins (1,422+1,070-446=2,046). Isn't it?

A. Indeed, this is an excellent comment and we have incorporated it in the new section "Conclusion" requested by Reviewer No. 2, on lines 432-434: "Furthermore, a combined strategy using both types of libraries will increase the probability of obtaining a more comprehensive coverage of transcripts to enhance the corresponding application to which the libraries are to be used."

Reviewer 2 Report

The present manuscript is originally, interesting article, which correspond with the aims and scope of the journal Microorganisms. The mechanisms underlying Symbiodiniaceae-cnidarian mutualistic symbiosis are still poorly understood even though it is a fundamental association for the survival and biodiversity of coral reefs. In view of that the authors present important information concerning developing a molecular tool which allow progress in the acquisition of a major understanding of the molecular events underlying the biology, metabolism and symbiosis of Symbiodiniaceae.   The advantage of the Dino-SL sequence found in the Symbiodiniaceae transcripts to generate a cDNA library from S. microadriaticum was proved in comparison with a SMART technology generated library.

The Title is clear and informative and in unison with the contents of the article.

The article is clearly laid out and all elements as abstract, introduction, materials and methods, results and discussion are presented.

 Abstract is clearly written and summarize the important results and conclusions of the paper.

Introduction: The authors summarized and described the previous surveys based the progress in the study of cnidarian-dinoflagellate molecular and functional genomic tools applied to corals and Symbiodiniaceae. The Dino-SL strategy provides a molecular tool to obtain high throughput cDNA libraries.

In the end of the manuscript the aim of the investigation have to specified.

Material and Methods: covered the comprehensively information about the cell lines and cultures,  RNA Isolation, cDNA synthesis and amplification, generation of cDNA libraries and sequencing, Bioinformatic and Analysis Y2HS Screening.

The Results are clearly presented and illustrated with appropriate Figures and Tables. The Tables and Figures give useful information, which verify the analyzed data and conclusions. All cited in the manuscript references are correctly presented in the List of Referеnces.

Discussion is comprehensive, written based on detailed analyses of the results.

The conclusion should be specified separately at the end of the paper.

Author Response

Dear Reviewer No. 2:

Please find below our response to your queries:

Q. In the end of the manuscript the aim of the investigation have to specified.

A. We think you meant “end of the introduction”. We had already written the aim of the investigation (highlighted at the end of the introduction) but to make it more obvious, we added a sentence on lines 105-108: "In this work, in an attempt to obtain a library with higher complexity than the conventionally generated SMART libraries, we sought to generate a cDNA library of Symbiodinium microadriaticum ssp. microadriaticum using the Dino-SL strategy (DINO library) to amplify the dinoflagellate cDNA and clone it into the pGADT7-rec plasmid. Then..."

Q. The conclusion should be specified separately at the end of the paper.

A. We added the section “Conclusion” separately on lines 428-436:

"5. Conclusion

Altogether, our comparative analysis clearly indicates that using the Dino-SL sequence found in the transcripts of S. microadriaticum to generate cDNA libraries allows to obtain a higher transcript recovery when compared to a SMART technology generated library. Furthermore, a combined strategy using both types of libraries will increase the probability of obtaining a more comprehensive coverage of transcripts to enhance the corresponding application to which the libraries are to be used. This molecular tool will enable progress in the acquisition of a major understanding of the molecular events underlying the biology, metabolism and symbiosis of Symbiodiniaceae."